# Prevalence and Characterization of Beta-Lactam and Carbapenem-Resistant Bacteria Isolated from Organic Fresh Produce Retailed in Eastern Spain

**DOI:** 10.3390/antibiotics12020387

**Published:** 2023-02-14

**Authors:** Ana Isabel Jiménez-Belenguer, Maria Antonia Ferrús, Manuel Hernández, Jorge García-Hernández, Yolanda Moreno, María Ángeles Castillo

**Affiliations:** 1Centro Avanzado de Microbiología de Alimentos, Universitat Politècnica de València, C/Camí de Vera s/n, 46022 València, Spain; 2Instituto de Ingeniería del Agua y Medio Ambiente, Universitat Politècnica de València, C/Camí de Vera s/n, 46022 València, Spain

**Keywords:** organic vegetables, resistances, β-lactams, third-generation cephalosporins, carbapenemases, ESBL

## Abstract

Fresh fruits and vegetables are potential reservoirs for antimicrobial resistance determinants, but few studies have focused specifically on organic vegetables. The present study aimed to determine the presence of third-generation cephalosporin (3GC)- and carbapenem-resistant Gram-negative bacteria on fresh organic vegetables produced in the city of Valencia (Spain). Main expanded spectrum beta-lactamase (ESBL)- and carbapenemase-encoding genes were also detected in the isolates. One hundred and fifteen samples were analyzed using selective media supplemented with cefotaxime and meropenem. Resistance assays for twelve relevant antibiotics in medical use were performed using a disc diffusion test. A total of 161 isolates were tested. Overall, 33.5% presented multidrug resistance and 16.8% were resistant to all β-lactam antibiotics tested. Imipenem resistance was observed in 18% of isolates, and low resistance levels were found to ceftazidime and meropenem. Opportunistic pathogens such as *Acinetobacter baumannii*, *Enterobacter* spp., *Raoultella* sp., and *Stenotrophomonas maltophilia* were detected, all presenting high rates of resistance. PCR assays revealed *bla_VIM_* to be the most frequently isolated ESBL-encoding gene, followed by *bla_TEM_* and *bla_OXA-48_*. These results confirm the potential of fresh vegetables to act as reservoirs for 3GC- and carbapenem-producing ARB. Further studies must be carried out to determine the impact of raw organic food on the spread of AMRs into the community.

## 1. Introduction

Antimicrobial resistance (AMR) is currently one of the most threatening public health issues. According to the European Antimicrobial Resistance Surveillance Network, in 2020, more than 800 000 infections occurred in the European Union (EU) due to bacteria resistant to antibiotics, and more than 35,000 people died as a direct consequence of these infections [1]. In 2021, the most commonly reported bacterial species were *E. coli* (39.4% of all cases), followed by *S. aureus*, *K. pneumoniae*, *E. faecalis*, *E. faecium*, *P. aeruginosa*, *Acinetobacter* spp., and *S. pneumoniae* [2]. Between 2020 and 2021, the number of reported antibiotic-resistant bacteria (ARB) cases increased for all pathogens, but especially for *Acinetobacter* spp. (+43%), *E. faecium* (+21%), and *E. faecalis* (+14%).

Among antibiotic-resistant bacteria (ARB), the growing prevalence of Gram-negative extended spectrum ß-lactamase (ESBL)-producing bacteria resistant to third-generation cephalosporins (3GC) and carbapenems is a major concern worldwide, because these antibiotics are frequent last-resort treatment options. Infections with ESBL- and carbapenemase-producing pathogenic bacteria, mainly *Enterobacteriaceae*, cause high mortality and morbidity [2,3]. Moreover, their prevalence is increasing; in 2021, high percentages of carbapenem-resistant *Acinetobacter* were found in several countries, reaching 50% of clinical isolates [4,5,6,7]. The treatment options for infections with carbapenem-resistant bacteria are limited, especially because carbapenemase genes are usually localized on mobile genetic elements together with other genes, conferring resistance to other β-lactams, fluoroquinolones, and/or aminoglycosides. This also contributes significantly to their spread [8,9].

However, non-pathogenic ARB are also a matter of concern. Saprophytic Gram-negative species, whose reservoirs are soil and water, are an important source of antibiotic-resistant genes (ARG) in the environment, as they are often resistant to third-generation cephalosporins and carbapenems due to the presence of ESBL genes on their chromosomes or the carriage of plasmids containing ESBL genes [10]. 

Although the prevalence of ESBLs tends to differ among countries, TEM, SHV, CTX-M, and OXA-type enzymes are the main classes of ESBLs [11,12]. The most common carbapenemases found in Gram-negative bacteria from different sources belong to Ambler class A (*bla_KPC_*), class B metallo-b-lactamases (*bla_IMP_*, *bla_VIM_*, and *bla_NDM_*), and class D (*bla_OXA-48_*) β-lactamases [9].

The “One Health” concept focuses on the interconnected nature of human, animal, and environmental health and highlights zoonotic diseases, food safety, and antimicrobial resistance as three particularly relevant areas [13]. Considering that ARB can be found in samples from several different origins, including humans, wastewater, foods, animals, and environmental sources such as soil or water, it is widely accepted that the environment is a crucial point for the development and spread of antimicrobial resistance determinants [9,14,15]. Among environmental sources, fruits and vegetables are receiving increasing attention as transmission vehicles for ARB and ARG from the environment to humans because they are usually consumed fresh, without further processing. The presence of 3GC- and carbapenem-resistant bacteria in vegetables has been reported by several authors [10,11,16,17]. Fruits and vegetables can be contaminated by soil, fertilizers, and irrigation water or by cross-contamination during harvesting, distribution, and storage [17,18]; ARB can then be transferred through the food chain from primary production to the final consumer [19]. In this sense, the growing demand for fresh organic vegetables is a matter of concern. 

The term “organic agriculture” refers to agricultural methods that aim to limit environmental impact by using natural substances and processes, including organic fertilizers and a predominant reliance on ecosystem services and non-chemical measures for pest prevention and control [20,21]. Globally, organic production has experienced spectacular growth in recent years, and, despite the impact of the pandemic, the global market for organic food strongly increased in 2020, exceeding 120 billion euros [22]. In Comunitat Valenciana (Spain), there was an increase in organic food production of 81.2% between 2016 and 2020 [23]. 

Consumers are interested in organic vegetables because they subjectively understand that these are healthier and better-quality foods than conventional products. However, there is a knowledge gap about some aspects of organic food microbiological safety [21,24]. One question concerns the potential risk of organic vegetable food products transmitting antibiotic resistance determinants, bacteria (ARB), and genes (ARG). When the prevalence of ARB in organic agriculture is compared with that of conventionally grown vegetables, there is a lack of consensus, mainly due to the fact that very few studies have been performed on organic vegetables. Some authors have reported that the presence of ARB and ARG in soils is increased by animal manure application or the reuse of municipal wastewater for irrigation [25,26,27,28,29]. However, other studies have found no increase or even a lower abundance of antibiotic-resistant bacteria isolated from organic vegetables compared to that of conventionally grown produce [30,31,32,33,34]. It seems that, although the use of human and animal manure as fertilizer is undoubtfully a significant route for ARG to enter agricultural soils, when the appropriate delay or pre-treatment of manure is performed, the impact of biosolids or sewage sludge application does not affect the abundance of pathogenic bacteria or ARB on vegetables at harvest [27,35,36]. However, this is still a matter of controversy.

Thus, further information related to the potential health hazards of organic vegetable products is required as, in the absence of adequate comparative data, generalized conclusions cannot be achieved, and further efforts are needed to understand the risks of ARG spread and ARB infections associated with organic vegetables [37,38,39]. The present study aimed to investigate the prevalence of third-generation cephalosporine- and carbapenem-resistant Gram-negative bacteria on fresh organic fruits and vegetables retailed in the city of Valencia (Eastern Spain). The main ESBL- and carbapenemase-encoding genes of the isolates were also detected.

## 2. Results

### 2.1. Isolation and Identification of β-Lactamase-Producing Enterobacterales

After growing in mSuperCARBA and MacConkey media supplemented with cefotaxime, at least one colony was observed from each of the 115 (100%) analyzed vegetable samples. A total of 673 presumptive 3GC- and carbapenem-resistant colonies were obtained (20% from lettuce, 23% from spinach, 40% from cabbage, and 36.6% from strawberries), with 74% being oxidase-positive. Of these 673 colonies, 161 were randomly chosen for further analysis (confidence level > 95% [40]), maintaining both the rate of oxidase-positive/negative isolates and the percentages obtained from each type of vegetable (33 from lettuce, 52 from spinach, 59 from cabbage, and 17 from strawberries) (Table 1). 

Using the API system and 16 rRNA gene partial sequencing, 15 isolates from 12 (10.4%) samples were identified as *Acinetobacter* sp., with nine of them as *Acinetobacter baumannii*. Twenty-nine presumptive 3GC- and carbapenem-resistant *Enterobacteriaceae* were isolated from 15 of the 115 (13.04%) vegetable samples: *Enterobacter* (*n* = 10), *Pantoea* sp. (*n* = 6), *Rahnella* sp. (*n* = 5), *Raoutella* sp. (*n* = 3), *Serratia* sp. (*n* = 2), *Leclercia* sp. (*n* = 1), *Buttiauxella agrestis* (*n* = 1), and *Proteus* sp (*n* = 1). The other 117 isolates were non-fermenter and oxidase-positive bacteria: *Pseudomonas* spp. (*n* = 72, where one isolate was *P. aeruginosa*, *Stenotrophomonas* sp. (*n* = 28, mainly *S. maltophilia* (*n* = 12)), *Burkholderia* sp. (*n* = 5), *Elizabethkingia* sp. (*n* = 2), *Ralstonia* sp. (*n* = 3), *Pasteurella* sp. (*n* = 3), *Sphingobacterium multivorum* (*n* = 2), *Achromobacter xylosidans* (*n* = 1), and *Ochrobactrum anthropi* (*n* = 1) (Appendix A).

### 2.2. Antibiotic Susceptibility Patterns

From the results of the disc diffusion test, 157 strains (97.51%) were resistant to at least one antibiotic. Only one *Stenotrophomonas* isolate was susceptible to all antibiotics tested. Three *Pseudomonas* isolates presented intermediate susceptibility to amoxicillin (AMC) and were susceptible to the rest of antibiotics. Total resistance levels to AMC and ampicillin (AMP) reached 91.9% and 95.7%, respectively (Table 2). 

The results obtained for different antibiotic classes are shown in Figure 1. All (100%) *Enterobacteriaceae*, 80% of *Acinetobacter*, 90.3% of *Pseudomonas*, and 89.3% of *Stenotrophomonas* isolates were resistant to AMP. *Enterobacteriaceae*, *Pseudomonas* sp., and *Stenotrophomonas* sp. isolates showed 85.7%, 99%, and 96% resistance to this antibiotic, respectively.

Among the *Enterobacteriaceae* isolates, five were resistant only to AMP and the other five only to AMP and AMC. For carbapenems, only two *Enterobacter cloacae* isolates were resistant to imipenem (IPM), and one of them was also resistant to meropenem (MEM). One *Routella* strain was shown to be resistant to imipenem but not to meropenem. 

All *Acinetobacter* isolates (100%) showed resistance to at least three antibiotics, including AMC. Two *A. baumannii* isolates were resistant to imipenem and meropenem, while the other was only resistant to imipenem.

Regarding third-generation cephalosporins, 116 isolates (72%) showed resistance to at least one of the antibiotics tested. Overall, 61.5% of isolates were resistant to cefotaxime (CTX), 43.5% to ceftriaxone (CRO), and 21.1% to ceftazidime (CAZ). Resistance to cefotaxime was noted for more than 60% of *Acinetobacter*, *Pseudomonas*, and *Stenotrophomonas* isolates. CRO resistance levels in *Acinetobacter* (73%) and *Stenotrophomonas* (67.8%) were shown to be statistically significant (χ^2^ = 18.026, *p* = 0.0012).

Thirty isolates (18.6%) showed resistance to carbapenems: 15.5% to IMP and 12.4% to MEM. Highly significant IMP resistance levels were observed for *Stenotrophomonas* (42.8%) (χ^2^ = 18.026, *p* = 0.0012).

Resistance to non-β-lactam antibiotics was also studied. The most prevalent resistance was against quinolones (27 (16.1%) isolates resistant to nalidixic acid (NA)), followed by tetracycline (TE) (19 (12.4%) isolates) and gentamicin (CN) (17 (10.6%) isolates). Resistance to the fluoroquinolones ciprofloxacin (CIP) and levofloxacin (LEV) was observed for 12 (7.5%) and three (1.9%) isolates, respectively. 

All *Enterobacteriaceae* were susceptible to CIP, LEV, and TE, while only two isolates showed resistance to CN and three to NA. The 15 *Acinetobacter* isolates were susceptible to ciprofloxacin. However, resistance rates to NA (46.7%), LEV (three isolates (20%), two *A. baumannii* and one *A. calcoaceticus*), and CN (four isolates (26.7%), two being *A. baumannii*) were significantly higher for *Acinetobacter* than those observed in the rest of bacterial groups (χ^2^ = 13.671, *p* = 0.0084; χ^2^ = 29.754, *p* = 0.0000; and χ^2^ = 11.088, *p* = 0.0256, respectively). 

No significant relationship was found among resistance to any class of antibiotic and the origin of the isolate, except for resistance to CAZ and spinach (χ^2^ = 9.675, *p* = 0.0466). However, among the 59 isolates obtained from cabbage, high levels of resistance to CTX (*n* = 35) were observed. Similarly, 19 and 16 out of the 33 isolates from lettuce were resistant to CTX and CRO, respectively, while isolates from spinach also presented high levels of resistance to CRO (29 out of 52). 

Multidrug resistance (MDR) was detected in 36 (22.4%) of the 161 isolates: 11 (33.3%) from lettuce, 13 (25%) from spinach, and 12 (20.3%) from cabbage). No MDR was detected in strawberry isolates (Table 3). 

Twenty-three MDR isolates showed resistance to 3G-cephaloporins: one *Enterobacter* sp., one *Elizabethkingia* sp., one *Sphingebacterium* sp., two *Pasteurella* spp., four *Burkholderia cepacia*., 14 *Stenotrophomonas* spp., eight *Pseudomonas* spp. and four *Acinetobacter* spp. All of these isolates were also resistant to carbapenems, except for one *Sphingebacterium*, three *Stenotrophomonas*, one *Acinetobacter* sp., and three *Pseudomonas* spp. isolates. 

Two isolates of *Acinetobacter*, *2 A. baumannii* and *2 A. calcoaceticus*, presented MDR patterns (26.7% of all isolates belonging to this genus). The two *A. baumannii* were resistant to all antibiotic classes except tetracycline. The only *P. aeruginosa* strain detected in the samples was resistant to penicillins, 3G-cephalosporins, and tetracycline. One *Pseudomonas putida* strain, obtained from a spinach sample, was resistant to all types of antibiotics tested. Finally, only one *Enterobacteriaceae* (*Enterobacter* sp.) presented an MDR profile.

When comparing the performance of the two selective media used for the isolation of resistant bacteria, similar rates were obtained between MacConkey supplemented with cefotaxime and mSuperCARBA for all the antibiotics tested, except for IMP- and MEM-resistant isolates, which were isolated mainly from the mSuperCARBA medium (93.3%).

### 2.3. Antibiotic Resistance Genotype Profile

Genes encoding β-lactamases were detected in 64.6% (104/161) of 3GC- and carbapenem-resistant isolates obtained from fresh produce. The most frequently detected gene was *bla_VIM_* (*n* = 43, 26.7%), followed by *bla_IMP_* (*n* = 38, 23.6%), *bla_TEM_* (*n* = 33, 20.5%), *bla_OXA-48_* (*n* = 28, 17.4%), *bla_SHV_* (*n* = 20, 12.4%), *bla_CMY-2_* (*n* = 13, 8.1%), and *bla_KPC_* (*n* = 7, 4.3%) (Table 4).

Overall, 53 isolates (33%) harbored more than one gene. For *Acinetobacter baumannii*, at least one gene was detected in six out of nine isolates (66.6%): one gene was detected in three isolates (*bla_SHV_*, *bla_VIM_*, and *bla_TEM_*), one isolate harbored two resistance genes (*bla_IMP_*-*bla_OXA-48_*), and two isolates from two different samples carried three genes (*bla_TEM_*-*bla_CMY-2_*-*bla_KPC_*). 

Out of 29 *Enterobacteriaceae* isolates, 21 (*Enterobacter* sp., *n* = 4; *Serratia* sp., *n* = 2; *Proteus* sp., *n* = 1; *Rahnella* sp., *n* = 5; *Pantoea* sp., *n* = 5; *Buttiauxella agrestis*, *n* = 1; and *Raoutella* sp., *n* = 3) carried resistance genes: one gene was detected in nine isolates (*bla_OXA-48_*, *n* = 5; *bla_SHV_*, *n* = 2; *bla_IMP_*, *n* = 1; and *bla_VIM_*, *n* = 1), seven isolates harbored two genes (*bla_TEM_*-*bla_OXA-48_*, *n* = 3; *bla_TEM_-bla_VIM_*, *n* = 2; *bla_SHV_*-*bla_CMY-2_*, *n* = 1; and *bla_IMP_*-*bla_OXA-48_*, *n* = 1), and three genes were detected in five isolates (*bla_SHV_**-bla_IMP_-bla_VIM_*, *n* = 1; *bla_TEM_-bla_VIM_-bla_IMP_*, *n* = 1; *bla_TEM_*-*bla_SHV_*-*bla_VIM_*, *n* = 2; and *bla_SHV_*-*bla_OXA-48_*-*bla_VIM_*, *n* = 1).

For *Stenotrophomonas* (28 isolates), no genes were detected in four isolates; one gene was detected in two isolates (*bla_IMP_* and *bla_OXA-48_*); 14 isolates carried two genes (*bla_IMP_*-*bla_OXA-48_*, *n* = 8; *bla_VIM_-bla_IMP_*, *n*= 4; *bla_SHV_**-bla_IMP_*, *n* = 1; and *bla_VIM_*-*bla_CMY-2_*, *n* = 1), and different combinations of three genes were detected in eight isolates (*bla_TEM_*-*bla_CMY-2_*-*bla_VIM_*, *n* = 4; *bla_SHV_*-*bla_OXA-48_*-*bla_IMP_*, *n* = 3; and *bla_TEM_-bla_VIM_-bla_IMP_*, *n* = 1). 

Finally, 24 out of 89 isolates of *Pseudomonas* sp. and other non-fermenting bacteria carried one resistance gene (*bla_TEM_*, *n* = 7; *bla_SHV_*, *n* = 1; *bla_IMP_*, *n* = 11; *bla_CMY-2_*, *n* = 2; *bla_VIM_*, *n* = 3; and *bla_KPC_*, *n* = 2), while 16 isolates harbored two genes (*bla_TEM_*-*bla_SHV_*, *n* = 2; *bla_TEM_*-*bla_CMY-2_*, *n* = 2; *bla_TEM_-bla_IMP_*, *n* = 1; *bla_TEM_*-*bla_KPC_*, *n* = 3; *bla_VIM_*-*bla_CMY-2_*, *n* = 1; and *bla_VIM_-bla_IMP_*, *n* = 7), and four carried three genes (*bla_SHV_*-*bla_IMP_*-*bla_OXA-48_*, *n* = 3 and *bla_TEM_*-*bla_VIM_*-*bla_CMY-2_*, *n* = 1).

The statistical analysis showed that the presence of *bla_OXA-48_* in *Stenotrophomonas* and *Enterobacteria* was significantly greater than in the rest of the bacterial groups (χ^2^ = 37.933, *p* = 0.0000). The presence of *bla_VIM_* in *Stenotrophomonas* was also significantly higher than in the rest of bacterial groups (χ^2^ = 24.365, *p* = 0.0001).

Among the 27 isolates that showed resistance to all the β-lactam antibiotics in the disc diffusion assay, 16 harbored at least one carbapenemase-encoding gene. In three isolates, only the *bla_TEM_* gene was detected, while in eight isolates, we did not detect any of the tested genes. 

The statistical correlation between the presence of each gene and the origin of the isolate was investigated: *bla_TEM_* was significantly more frequently detected in cabbage (χ^2^ = 10.405, *p* = 0.0154), *bla_KPC_* in strawberries (χ^2^ = 18.981, *p* = 0.0003), *bla_OXA-48_* in lettuce (χ^2^ = 12.893, *p* = 0.0049), and *bla_SHV_*, *bla_IMP_*, and *bla_VIM_* in spinach (χ^2^ = 19.699, *p* = 0.002, χ^2^ = 19.358 and 20.359, *p* = 0.0002 and 0.0001, respectively). No correlation was found for the *bla_CMY-2_* gene.

## 3. Discussion

The presence of antibiotic-resistant bacteria on fresh organic vegetables is a growing public health concern, due to an increase in the consumption of these products in recent years and evidence that fresh vegetables constitute a source of ARB, in addition to representing a possible route for the dissemination of resistance genes in the community and the environment [34,41]. However, very few studies are available on the actual food safety and environmental risks that occur due to the presence of antibiotic resistance determinants in fresh organic vegetables [16].

Thus, the present study aimed to detect ARB and ARG in organic vegetables grown and distributed in Valencia (Spain) that are usually consumed raw. We focused on Gram-negative bacteria because a large proportion of infections caused by drug-resistant microorganisms in community and healthcare settings are recognized to be caused by these types of bacteria [42]. Due to their special significance in human medicine, ESBL- and carbapenem-producing Gram-negative bacteria were specifically detected. 

A total of 115 organic vegetable samples (lettuce, spinach, cabbage, and strawberries) were analyzed. These types of vegetables were selected because they grow near the soil and are usually eaten uncooked. The search for ARB was carried out using selective enrichment and culture media supplemented with β-lactam antibiotics. All the samples were contaminated with ARB, and more than 97% of the 161 selected isolates presented resistance to at least one antibiotic in the disc diffusion test. Moreover, 22.4% presented multidrug resistance and 14.9% were resistant to all β-lactam antibiotics tested, forming 18% of all isolates resistant to carbapenems. Nevertheless, the prevalence of resistance to CAZ and MEM was low among all the isolated bacteria. Among non-β-lactam antibiotics, resistance ranged from 16% for NA to 2% for LE. 

Overall, these resistance rates confirm the potential of fresh vegetables to act as a reservoir for AR determinants in the environment. Additionally, the consumption of these raw vegetables may result in the transmission of resistance to other commensal or pathogenic gut microbiota through gene transfer mechanisms, a process previously described as a “silent food safety concern” [34,43], contributing to the dissemination of antibiotic resistance within the community.

In this sense, one of the major threats is the growing prevalence of Enterobacteriaceae resistant to third-generation cephalosporins (3GC) and carbapenemases [44]. In this study, we detected 3GC-resistant Enterobacteriaceae in 13% of the samples; *Enterobacter* was the most prevalent genus, followed by *Pantoea* and *Rahnella.* Our results are similar to those of other studies on conventional vegetables, in which *Enterobacter* was the most frequently detected genus, followed by *Pantoea*, *Serratia*, *Citrobacter*, and *Rahnella* [10]. In addition, in a study performed in the same geographical area [34], *Pantoea Enterobacter* and *Serratia* were the predominant Enterobacteriaceae genera in organic samples. 

Regarding the numbers of ESBL-producing Enterobacteriaceae isolated from raw vegetables, it is difficult to compare our findings with those of other studies due to significant differences in study design, sampling strategy, number, and type of vegetables analyzed, isolation methodologies, and geographical sampling zone. This lack of homogeneity has been also pointed out by other authors [16,31]. However, our results (13%) cannot be considered high and are in the range of those obtained by Reuland et al. [45] (6%), Raphael et al. [42] (12%), Moon et al. [46] (9.09%), and Ritcher et al. [47] (17.4%) for conventional and organic vegetables under different experimental conditions. On the contrary, some authors have found higher rates on conventional vegetables, from 25.4% [41] to 40% [25,27]. Overall, organic vegetables analyzed in this study did not present higher contamination rates than those found by other authors on conventional non-organic produce.

Among the Enterobacteriaceae isolated from organic vegetables, resistance levels to CRO were below 60%. Moreover, only one strain presented an MDR profile. Resistance to carbapenems and CN was also low, reaching 46% for MDR and 70 to 90% for aminoglycoside resistances, far from that obtained by other authors on non-organic vegetables [11,47]. The presence of one MDR, including all carbapenems, *Enterobacter* strain, is coincident with other studies [16], and may represent a public health concern for immunocompromised patients.

It is also remarkable that we found one *Raoultella* isolate resistant to IPM. *Raoultella* organisms are often misidentified, due to their close relatedness to *Klebsiella* spp. They are commonly found in the natural environment and for many years were considered saprophytic bacteria. However, in recent years, infections in immunocompromised patients have been frequently reported, with high mortality rates. In addition, MDR isolates, including carbapenemase-producing strains, have been described both in clinical settings [48] and in the environment [49].

*Acinetobacter* isolates were isolated from 9.5% of the analyzed samples, whereas *A. baumannii* was present in nine samples (5.6%). These rates are similar to those found in conventional non-organic vegetables and fruits [50]. This species is an important hospital-acquired pathogen worldwide, widely found in intensive care units where it can cause severe infections. Moreover, there are increasing reports of multidrug-resistant non-baumannii Acinetobacter that cause infections in healthcare facilities around the world [5,6,51]. *Acinetobacter* are intrinsically resistant to penicillins and generally present multidrug resistance to third-generation cephalosporins, aminoglycosides, and fluoroquinolones. Their antibiotic-resistant genes are usually harbored in class 1 integrons, which has allowed them to become a major contributor to resistance among clinical and environmental Gram-negative bacteria [51,52]. In this study, we observed worrying resistance rates: 26.7% of *Acinetobacter* spp. presented MDR patterns, and two *A. baumannii* isolates were resistant to all antibiotic classes except tetracycline. High levels of antibiotic resistance in *Acinetobacter* present in conventional vegetables have been previously detected by other authors [5,25,30,53].

Most (73%) of the 161 Gram-negative isolates were oxidase-positive, non-fermenter bacteria, including a high percentage of *Pseudomonas* and *Stenotrophomonas*. In contrast to other studies [42,54], most of the isolates were saprophytic, and only one *P. aeruginosa* isolate was detected. Nonetheless, resistance rates were found to be high, and one *Pseudomonas putida* isolate was resistant to all antibiotics tested.

Regarding the *Stenotrophomonas* isolates, they all presented high rates of resistance to CRO and IPM, and almost 40% were resistant to all β-lactam antibiotics tested. *Stenotrophomonas* are environmental bacteria and are considered to be emergent pathogens, mainly producing severe pulmonary and bloodstream infections in vulnerable populations with mortality rates from 24% to 58%. *S. maltophilia* was recently described as the most common Gram-negative carbapenem-resistant pathogen isolated from bloodstream infections acquired in community and hospital settings in the USA [55]. Therefore, its presence in vegetables and the high levels of resistance detected cannot be underestimated.

Genotypic characterization using PCR assays revealed *bla_VIM_* to be the most frequently isolated carbapenemase-encoding gene, followed by *bla_TEM_* and *bla_OXA-48_*. Previous studies have also isolated carbapenemase-producing *Enterobacteriaceae* from fresh vegetables [11,17,56] and pointed out both the great spread of carbapenemase-encoding genes in the environment and the risk of acquisition for this type of ARG through the consumption of fresh vegetables. 

The high prevalence of *bla_OXA-48_* in our samples differs from the results of previous studies from other countries [11,17]. Carbapenem-resistant OXA-48 genes are a significant cause of carbapenem resistance, and the emergence of OXA enzymes, particularly in *A. baumannii*, has transformed these lactamases into a major problem, diminishing the clinical efficacy of carbapenems [57]. 

Regarding the presence of pathogenic bacteria in the analyzed fruits and vegetables, a relevant result of this study is that we found exclusively opportunistic pathogens. Strict farming regulations in most European and North American countries are aimed at protecting vegetables from contamination with pathogenic microorganisms and, consequently, most antibiotic- and multidrug-resistant bacteria are expected to be predominantly saprophytic and opportunistic organisms [41,58].

Our results confirm that organic vegetables are safe and do not constitute a direct threat to immunocompetent consumers. However, the considerable number of MDR opportunistic bacteria raises concerns about the consumption of this type of food by immunocompromised patients. On the other hand, the fact that all the vegetable samples contained at least one colony of resistant bacteria confirms the role of these foods as reservoirs of antibiotic resistance genes. Saprophytes in fresh produce have been found to harbor drug resistance genes that are also found in internationally circulating strains of pathogens; they may thus serve as a reservoir for drug resistance genes that enter pathogens and affect human health [42]. Further research should thus focus on clarifying and quantifying the involvement of saprophytic bacteria in ARG transfer to pathogens as well as in extraintestinal opportunistic infections after transmission by food [59].

Antimicrobial resistance surveillance programs primarily focus on food from animal origins. However, according to the present work and previous studies [16], monitoring antibiotic-resistant bacterial reservoirs in vegetables seems to be equally important. Some authors have even suggested that including the detection of antibiotic-resistant non-pathogenic Enterobacteriaceae species in fresh vegetables during epidemiological surveillance routines could help to quantify the dissemination of multidrug resistance in non-clinical environments and evaluate the potential role of the consumption of fresh vegetables in spreading resistance in the community [34].

## 4. Materials and Methods

### 4.1. Sampling

Between June 2021 and May 2022, a total of 115 fresh vegetable samples with the European organic label were purchased weekly in 12 supermarkets and popular markets in the city of Valencia, Spain. The vegetables included lettuce (*Lactuca sativa, n* = 30), spinach (*Spinacia oleracea, n* = 30), cabbage (*Brassica oleracea* var. *sabellica, n* = 15; *Brassica oleracea* var. *capitata f. rubra*, *n* = 15), and strawberries (*Fragaria, n* = 25). Each sample consisted of a batch of 3–4 pieces for leaf vegetables and 750 g for strawberries. 

The vegetable samples were handled under aseptic conditions after their arrival at the laboratory. Unwashed leaves were cut with sterile scissors into small portions and the resulting fragments were mixed prior to homogenization. Strawberries were processed by discarding the sepal and pedicle, leaving only the consumable fruit. 

The samples were pre-enriched to isolate third-generation cephalosporin (3GC)- and carbapenem-resistant bacteria. From each sample, two portions of 10 g were aseptically placed in separate sterile flasks containing 90 mL of buffered peptone water (BPW) (Scharlau, Barcelona, Spain) with cefotaxime (2.5 mg/L) and 90 mL of trypticase soy broth (TSB, Oxoid, England) supplemented with meropenem (1.0 mg/L). Both broths were supplemented with vancomycin (5 mg/L) to ensure the inhibition of Gram-positive bacterial growth. The samples were homogenized using a Stomacher sample blender and incubated at 37 °C overnight. Thereafter, 50 mL of the sample suspension was centrifuged at 4000 rpm for 10 min for further DNA extraction [14]. For the detection of antibiotic-resistant bacteria, one loopful of each broth was streaked onto mSuperCARBA (CHROMagar™ Paris, France) and MacConkey agar supplemented with cefotaxime (2.5 mg/L), both prepared according to the manufacturer’s instructions. The plates were aerobically incubated for 24 h at 37 °C. 

### 4.2. Isolation and Characterization of Isolates

A maximum of 3 different suspected 3GC- and 3 carbapenem-resistant colonies were picked from each MacConkey and mSuperCARBA plate, respectively, based on the morphological and color-coded distinction of colonies according to the manufacturers’ protocols. Thus, 6 different isolates were recovered from each sample, purified, and identified using Gram, catalase, and oxidase tests. The isolates were stored at −80 °C until use.

The selection of colonies was performed, and isolates were identified using the API phenotypic identification system (API20E and API 20NE strips, BioMèriux, Marcy-l’Étoile, France). When the identity match with the database was less than 99%, identification was confirmed by the partial sequencing of the 16S rDNA gene to amplify the 16S rRNA gene (1500 bp) using the universal bacterial primers 27F (5′-AGAGTTTGATYMTGGCTCAG-3′) and 1492R (5′-GGTTACCTTGTTACGACTT-3′) for the *Enterobacteriaceae* isolates [60] and 27F and 1525R (5′-AGAAAGGAGGTGATCCAGCC-3′) for the presumptive *Acinetobacter* (non-fermenter strains) isolates [61]. The PCR conditions for *Enterobacteriaceae* were: one cycle at 95 °C for 5 min, 35 cycles comprising the denaturation phases at 95 °C for 1 min, annealing at 55 °C for 1 min, extension at 72 °C for 1 min, and a final extension cycle at 72 °C for 10 min. For presumptive *Acinetobacter* strains, the PCR conditions were as follows: one cycle at 95 °C for 2 min, 35 cycles at 94 °C for 30 s, 65 °C for 30 s, 72 °C for 2 min, and a final extension at 72 °C for 10 min. All the reactions were performed in a PTC-100 Thermal Cycler (MJ Research, St. Bruno, QC, Canada).

The PCR products were electrophoresed on a 2% agarose gel (Pronadisa, Madrid, Spain) with 100 mL of Tris-acetate-EDTA buffer and 5 μL of RedSafe™ solution (Intron Biotechnology, Washington, USA) and scanned with a gel monitoring system (Transilluminator Vilber, Lourmat, France). After gel electrophoresis, the amplicons were purified using the GenElute™ PCR Clean-Up Kit (Sigma-Aldrich, Madrid, Spain) following the manufacturer’s instructions. The representative 16S rDNA fragments of the different isolates were sequenced by cycle extension in an ABI 373 DNA sequencer (Applied Biosystems; Foster City, CA, USA). An approximately 800 bp sequence was obtained per fragment, which was compared with sequences deposited in the GenBank database using the online BLAST program (http://www.ncbi.nlm.nih.gov/BLAST/ accessed on 22 October 2022). 

### 4.3. Antibiotic Susceptibility Testing of Isolates 

The isolates were subjected to susceptibility testing using the disc diffusion method against 12 antibiotics (Antimicrobial Susceptibility Test Disc, OXOID Ltd., England, United Kingdom), namely ampicillin (AMP, 10 µg), amoxicillin (AMC, 20 µg), cefotaxime (CTX, 30 µg), ceftriaxone (CRO, 30 µg), ceftazidime (CAZ, 30 µg), imipenem (IPM, 10 µg), meropenem (MEM, 10 µg), nalidixic acid (NA, 30 µg), levofloxacin (LEV, 5 µg), ciprofloxacin (CIP, 5 µg), gentamicin (CN, 10 µg), and tetracycline (TE, 30 µg), according to the CLSI guidelines [62]. 

*E. coli* ATCC 25922 and *Pseudomonas aeruginosa* ATCC 27853 were used as control strains. Multidrug resistance was defined as nonsusceptibility to at least three antibiotic classes [11,63]. Due to the existence of intrinsic resistance to AMP and AMC in *Pseudomonas* and *Acinetobacter* genus, MDR was considered when the isolates presented resistance to 3 classes of antibiotics in addition to penicillins. The results were interpreted according to the European Committee on Antimicrobial Susceptibility Testing [64] and isolates were classified as Susceptible or Resistant [65].

### 4.4. Antibiotic Resistance Gene Detection

DNA was extracted from all the selected isolates using the GenEluteTM Bacterial Genomic DNA Kit (Sigma-Aldrich, Madrid, Spain), according to the manufacturer’s instructions, and then screened for ESBL genes (*bla_TEM_*, *bla_SHV_*, and *bla_CYM-2_*) and carbapenemase-encoding genes (*bla_IMP_*, *bla_VIM_*, *bla_KPC_*, and *bla_OXA-48_*) using the primers described in Table 5. 

Three multiplex PCR (mPCR) reactions were performed for detecting ESBL, AmpC, and carbapenemase genes: (1) for the detection of *bla_SHV_*, *bla_TEM_*, and *bla_CMY-2_*; (2) for the detection of *bla_KPC_* and *bla_VIM_*; and (3) for the detection of *bla_IPM_* and *bla_OXA-48_*. These genes were selected because of their great prevalence among ESBL- and carbapenemase-producing Gram-negative pathogenic and environmental bacteria in Spain [66,67,68].

The mix for mPCR (1) included 2.5 μL template DNA; 2.5 mM MgCl_2_; 0.4 μM of each primer for *bla_SHV_*, and 0.2 μM for *bla_TEM_* and *bla_CMY-2_*; 0.2 mM of each dNTP; 5 U of Taq polymerase; and 1X PCR buffer, reaching a final reaction volume of 25 μL. The thermal cycler conditions for this assay were as described by [69]: 15 min at 94 °C, 30 cycles of amplification consisting of 1 min at 94 °C, 1 min at 55 °C, and 1 min at 72 °C, with 10 min for the final extension at 72 °C. As positive controls, we used *Klebsiella pneumoniae* subsp. *pneumoniae* ATCC 700603 for *bla_SHV_*, *Escherichia coli* ATCC 35218 for *bla_TEM_*, and our own positive *E. coli* strain (M1A mec8) for *bla_CMY-2_*. MilliQ water was used as a negative control. 

For mPCR (2) and (3), the mix consisted of 1X PCR buffer; 1.5 mM MgCl_2_; 0.4 μM of each primer; 0.125 mM of dNTPs; 2 U of Taq polymerase; and 2.5 μL of extracted DNA, reaching a final reaction volume of 25 μL. Amplification was carried out according to Poirel et al. [70] with slight modifications: 10 min at 94 °C and 36 cycles of amplification consisting of 30 s at 94 °C, 40 s at 55 °C, and 50 s at 72 °C, finishing with 5 min at 72 °C for the final extension. As positive controls, *K. pneumoniae* NCTC 13442, *K. pneumoniae* NCTC 13438, *K. pneumoniae* NCTC 13440, and *E. coli* NCTC 13476 were used for *bla_OXA-48_*, *bla_KPC_*, *bla_VIM_*, and *bla_IMP_*, respectively. MilliQ water was used as a negative control. 

All amplifications were developed in a Mastercycler^®^Pro (Eppendorf, Hamburg, Germany) thermal cycler. PCR products were detected by electrophoresis on 1.2% (*w*/*v*) agarose gel in TAE 1X (Tris 40 mM, acetic acid 20 mM, EDTA 1 mM) buffer with RedSafeTM (iNtRON Biotechnology, Sungnam, Korea) at 90 V for about 75 min and visualized under UV light. 

**Table 5 antibiotics-12-00387-t005:** PCR primer sequences, targets, and conditions.

Target Gene	Sequence	Conditions	Reference
*bla_TEM_*	5′-TTAACTGGCGAACTACTTAC-3′ 5′-GTCTATTTCGTTCATCCATA-3′	94 °C 15 min (1 cycle); 94 °C 1 min, 55 °C 1 min (36 cycles); 72 °C 1 min; 72 °C 10 min (elongation)	Kozak et al. (2009) [69]
*bla_SHV_*	5′-AGGATTGACTGCCTTTTTG-3′ 5′-ATTTGCTGATTTCGCTCG-3′		
*bla_CMY-2_*	5′-GACAGCCTCTTTCTCCACA-3′ 5′-TGGACACGAAGGCTACGTA-3′		
*bla_KPC_*	5′-CGTCTAGTTCTGCTGTCTTG-3′ 5′-CTTGTCATCCTTGTTAGGCG-3′	94 °C 10 min (1 cycle); 94 °C 30 s, 55 °C 40 s (36 cycles); 72 °C 55 s; 72 °C 5 min (elongation)	Poirel et al. (2011) [70]
*bla_OXA-48_*	5′-GCGTGGTTAAGGATGAACAC-3′ 5′-CATCAAGTTCAACCCAACCG-3′		
*bla_IMP_*	5′-GGAATAGAGTGGCTTAAYTCTC-3′ 5′-GGTTTAAYAAAACAACCACC-3′		
*bla_VIM_*	5′-GATGGTGTTTGGTCGCATA-3′ 5′-CGAATGCGCAGCACCAG-3′		

### 4.5. Statistical Analysis

Statistical analysis was performed using the Statgraphics (Centurion XVII) software (Statpoint Technologies, Inc. Warrenton, VA, USA). Antibiotic resistance and ARG detection were analyzed via a χ^2^ test, using contingency tables, to establish any possible dependent correlation with vegetable type and/or bacterial group. A probability value of less than 5% was considered significant.

## 5. Conclusions

The results of this study demonstrated a high abundance of relatively non-pathogenic ESBL- and carbapenemase-producing Gram-negative bacteria on fresh organic vegetables retailed in Valencia, Spain. Our results indicate that organic vegetables represent reservoirs and possible routes for the spread of ARB and ARG into the community. Although the absence of enteric pathogens was encouraging, saprophytic bacteria can carry antibiotic resistance determinants to consumers via the food chain and transfer them to gut commensal or pathogenic microorganisms. Moreover, the prevalence of opportunistic bacteria presenting high resistance levels was relevant in all the analyzed vegetables. Exposure to these opportunistic pathogens may result in the colonization of immunocompromised patients, causing severe and difficult-to-treat infections. Thus, the presence of ARB and ARGs on fresh organic vegetables that are consumed raw poses potential public health risks whose magnitude has yet to be established.

Our study points out the need for more research to determine the actual impact of raw organic food on the spread of ARGs, to understand the effects of human colonization with ARBs originating from fresh vegetables, and to qualitatively and quantitatively assess the health risks associated with the ARB/ARGs in these foods.

## Figures and Tables

**Figure 1 antibiotics-12-00387-f001:**
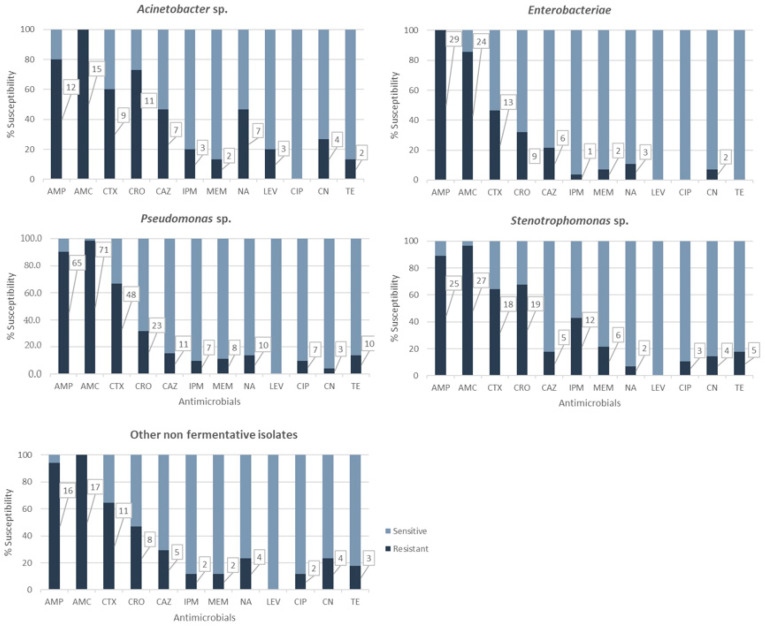
Antimicrobial resistance for each bacterial group; denotes significant differences between bacterial groups. Numbers in boxes correspond to number of resistant isolates.

**Table 1 antibiotics-12-00387-t001:** Representative bacterial species isolated from different fresh vegetables.

	Number of Isolates				
Identification	Lettuce	Spinach	Cabbage	Strawberry	Total
*Enterobacteriaceae*	6	15	4	4	29
*Acinetobacter* sp.	5	5	3	2	15
*Non-fermenters*	22	32	52	11	117
Total	33 (20%)	52 (32%)	59 (26.6%)	17 (11.1%)	

**Table 2 antibiotics-12-00387-t002:** Incidence of resistance to each antimicrobial compound in the isolates.

		Number of Resistant Isolates											
Vegetable	No. Tested Isolates	AMP	AMC	CTX	CRO	CAZ	IPM	MEM	NA	LEV	CIP	CN	TE
Lettuce	33	29	29	19	16	2	7	1	7	1	3	1	1
Spinach	52	51	50	36	29	15	11	8	7	0	3	7	11
Cabbage	59	53	58	35	18	11	7	6	13	2	4	8	5
Strawberries	17	15	17	9	7	6	1	4	0	0	2	1	2
Total	161	148	154	99	70	34	26	19	27	3	12	17	19
% of resistant isolates		91.9	95.7	61.5	43.5	21.1	16.1	11.8	16.8	1.9	7.5	10.6	11.8

AMP—ampicillin; AMC—amoxicillin; CTX—cefotaxime; CRO—ceftriaxone; CAZ—ceftazidime IPM—imipenem; MEM—meropenem; NA—nalidixic acid; CIP—ciprofloxacin; LEV—levofloxacin; CN—gentamicin; TE—tetracycline.

**Table 3 antibiotics-12-00387-t003:** Multidrug resistance patterns among isolates.

				No. Isolates (%)				
	Vegetables			Bacterial Groups				
Patterns	Lettuce (*n* = 33)	Spinach (*n* = 52)	Cabbage (*n* = 59)	*Acinetobacter* spp. (*n* = 15)	*Enterobacteriaceae* (*n* = 29)	*Pseudomonas* spp. (*n =* 72)	*Stenotrophomonas* spp. (*n = 28*)	*Other Non-Fermentative* (*n =* 17)
P-CP-CB	5 (15.1)	3 (5.7)			1 (3.4)		7 (25.0)	
P-CP-Q	4 (12.1)	3 (5.7)					3 (10.7)	4 (23.5)
P-CP-CN			1 (1.7)					1 (5.9)
P-CP-TE			1 (1.7)					1 (5.9)
P-Q-TE		1 (1.9)					1 (3.6)	
P-CP-CB-CN			1 (1.7)			1 (1.4)		
P-CP-CB-Q			1 (1.7)			1 (1.4)		
P-CP-Q-CN			1 (1.7)	1 (6.7)				
P-CP-Q-TE			3 (5.1)			3 (4.2)		
P-CP-CB-CN-TE		5 (9.6)	1 (1.7)			1 (1.4)	3 (10.7)	2 (11.8)
P-CP-CB-Q-CN	1 (3.0)		3 (5.1)	3 (20.0)			1 (3.6)	
P-CP-CB-Q-TE	1 (3.0)					1 (1.4)		
P-CP-CB-Q-CN-TE		1 (1.9)				1 (1.4)		
Total	11 (33.3)	13 (25)	12 (20.3)	4 (26.7)	1 (3.4)	8 (11.1)	15 (53.6)	8 (47.1)

P—penicillins; CP—third-generation cephalosporins; CB—carbapenems; Q—quinolones; CN—gentamicin; TE—tetracycline.

**Table 4 antibiotics-12-00387-t004:** Frequency of ARG detection in the isolates.

	Number of Isolates						
Vegetable	*bla_TEM_*	*bla_SHV_*	*bla_CMY-2_*	*bla_KPC_*	*bla_OXA-48_*	*bla_IMP_*	*bla_VIM_*
Lettuce	3	0	0	0	12	8	14
Spinach	9	15	5	0	9	22	21
Cabbage	13	4	5	3	4	4	8
Strawberries	8	1	3	4	3	4	0
Total (%)	33 (20.5)	20 (12.4)	13 (8.1)	7 (4.3)	28 (17.4)	38 (23.6)	43 (26.7)

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
