# Peer review of "Prevalence and Characterization of Beta-Lactam and Carbapenem-Resistant Bacteria Isolated from Organic Fresh Produce Retailed in Eastern Spain"

_antibiotics, 2023, doi:10.3390/antibiotics12020387_

Round 1

Reviewer 1 Report

The topic of this interesting manuscript is very important, since more and more studies are reporting on the role of vegetables and fruits to transmit pathogens and antibiotic resistance. 

Important comments and questions

1.     The title must be revised: First, “beta-lactam and carbapenems resistant bacteria” should be changed to “beta-lactam and carbapenem-resistant bacteria”. Second: 

2.     Strawberries is a fruit not vegetables. Therefore, the title must be change to reflect the type of samples analyzed in this study. 

3.     Why the authors did not take all the presumptive colonies for further analysis?

4.     Why the authors chose the antibiotics from CLSI guidelines and interpreted the results based on EUCAST guidelines? 

5.     The authors should compare their results with the studies on non-organic raw vegetables to show the differences between them in the prevalence of the mentioned bacteria. 

6.     The authors should add a table showing the 16S rRNA sequencing and API system results. Specifically, this table should include the isolates information and the results of BLASTing the sequences. This table could be added as a supplementary table.

7.     The authors should be consistent in their usage of the “isolates vs strains” ; “spp. vs sp.” ; “Antimicrobial vs antibiotic”, as well as all abbreviations.

8.     The authors should be careful in defining MDR isolates. For example, penicillins (AMP and AMC) are not used for as one of the antimicrobial categories and agents used to define MDR Pseudomonas aeruginosa and Acinetobacter spp. according to Magiorakos et al. (2012) https://www.sciencedirect.com/science/article/pii/S1198743X14616323

9.     On what basis did the author choose to screen for these genes: blaTEM, blaSHV blaCMY-2 blaKPC blaOXA-48 blaIMPblaVIM? Are these gene the most clinically relevant genes in Spain in general or Valencia city?

10.  Since no positive control strains were used for the detection of blaTEM, blaSHV  and blaKPC, did the authors at least sequenced the amplified products of these genes? 

11.  In the discussion, the authors state “A relevant result of this study is that we did not find any species potentially pathogenic for immunocompetent hosts”. The finding that some of the isolates are A. baumannii contradicts this statement. This pathogen is one of the leading causes of infections worldwide. 

12.  The manuscript needs major work in rendering the English language writing clear and concise. 

Please consider also the following minor corrections: 

Abstract

Line 15: “cephalosporines” should be corrected to “cephalosporins”

Line 16: “espectrum” should be corrected to “spectrum”

Line 17: “detalactamases” should be corrected to beta-lactamases”, add a hyphen between “carbapenemase” and “encoding” and “on” should be replaced by “in”

Line 18: Remove the coma and “, by”

Line 19: “eleven” should be corrected to “twelve” 

Line 22-23: delete “No pathogenic bacteria was isolated” 

Line 23-24: “baumanii” should be corrected to “baumannii

Line 26: “gen” should be corrected to “gene”

Line 27: “reservoir” should be corrected to “reservoirs”

Line 27:  “Further studies must be done”: its font size is smaller and must be adjusted

Line 28:  “ARGs” should be corrected to “AMRs”

Introduction

Line 39: Add a comma after “2021” and add a hyphen (-) between antibiotic and resistant 

Line 42: Add a hyphen (-) between “antibiotic” and “resistant” 

Line 43: “β-lactamases (ESBLs) producer” should be changed to “β-lactamases (ESBLs)-producing” 

Line 46: “carbapenemases producer” should be changed to “carbapenemases-producing”

Line 47: Add a comma after “increasing” 

Line 48: Add a hyphen (-) between “carbapenem” and “resistant” 

Lines 47-49: Add more references!

Line 50: “usually are” should be changed to “are usually”  

Line 53: “bacteria” should be removed 

Lines 54-55: Add a hyphen (-) between “antibiotic” and “resistant”

Line 60: Change “gram” to “Gram” Add a hyphen (-) between “Gram” and “negative”

Line 67: Add “the” before “environment” 

Line 70: Add “the” before “environment” 

Line 71: Change “process” to “processing”

Line 71: Add a hyphen between “carbapenem” and “resistant”

Lines 71-72: Add more references!

Line 74: Add a comma before “and ARB” 

Line 81: Add a comma after “years”

Line 83: Add a comma after “2020”

Line 89: change “antibiotic resistant” to “antibiotic-resistance”., delete “both” and add a semicolon after “determinants” 

Line 98: Add “the” before “impact” 

Line 99: Add “the” before “abundance” 

Line 102: Add “the” before “potential”

Line 107: delete “that are usually consumed raw,” 

Results

Line 115: change “, being 74% oxidase-positive” to “with 74% being oxidase-positive”

Line 115: change “By API system and sequencing” to “Using the API system and 16 rRNA gene partial sequencing”

Line 122: change “being 9 of them Acinetobacter baumanii” to “with 9 of them as Acinetobacter baumannii”

Lines 125-126: change “Other 117 strains were non-fermenter, oxidase positive, bacteria:” to “The other 117 strains were non-fermenter and oxidase positive bacteria, including”

Line 131: Change “By Disc Diffusion” to “By the disc diffusion”

Line 134: change “(AMC)” to “(AMP)”

Lines 159-161: Should be moved to be after line 143

Lines 187-188: The authors mentioned 2 MDR A. baumannii, however, in table 3 two are A. baumannii are listed as MDR with the P-CP-CB-Q- CN resistance profile. What is the identity of the 3rd isolate??

Tables and Figures

Table 1: In the title, add “Representative bacterial species ....” 

Table 1: “Strawberry” and “TOTAL” should be bold

Table 2: “CIP”, “CN,” and “TE” should be bold

Table 3:  Pseudomonas spp.”, “Stenotrophomonas maltophilia” and “Other non-fermentative” should be bold.

Table 3: The first column must have a header. Also, the numbers and the percentages in parenthesis should be indicated/labelled in the headers of the table. The Acinetobacter baumannii column should be changed to Acinetobacter spp. since the %s in the table are based on 15 isolates nit the 9 A. baumannii isolates. Same thing applies to S. maltophiliait should be replaced with Stenotrophomonas spp.

Table 4: “blaIMP” and “blaVIM” should be bold

Figure 1: Above each chart, the number of isolates from each bacterial group should be stated between parenthesis. 

Figure 1: the Y-axes and X-axes should be labeled with percentage and antibiotic, respectively. 

Discussion

Line 267: “disseminate” should be changed to “the dissemination of” 

Line 277: “ESBL producers” should be changed to “ESBL-producing” 

Line 279: Add “the” before “study” 

Line 299-300: “baumanii” should be corrected to “baumannii” 

Line 306: “baumanii” should be corrected to “baumannii

Line 308: “baumanii” should be corrected to “baumannii” and add a comma after “MEM”

Line 313: Add a comma after “saprophytic”

Line 328: Remove “both,” and add a hyphen between “carbapenemase” and “encoding” 

Line 332: Add a comma after “resistance” and “baumanii” should be corrected to “baumannii

Line 342: “foods” should be corrected to “food” 

Line 353: Add “the” before “dissemination” 

Line 355: Add “the” before “community” 

Materials and methods

Line 359: Replace “in” by “from”

Line 420: “Multiple resistance” should be changed to “Multidrug resistance” 

Conclusions

Line 459: Add “a” before “high” 

Line 460: “producer” should be changed to “producing” and “gram negative” to “Gram-negative” 

References

I recommend using the following references to expand the citation list and compare the outcomes of this study with other published data from similar studies. 

10.2217/fmb-2016-0082

10.1016/j.fm.2016.12.005

10.1111/1541-4337.12487

https://doi.org/10.1111/jam.15795

10.1128/AEM.03824-14

https://doi.org/10.1186/s40550-022-00092-7

10.4315/0362-028X.JFP-15-548

Author Response

Thank you

Reviewer 2 Report

The manuscript “Prevalence and characterization of beta-lactam and carbapenems resistant bacteria isolated from organic raw vegetables retailed in Eastern Spain” is not acceptable for publication due to the following reasons

There are many technical and language mistake  In Abstract “The present study aimed to determine the presence of 3rd generation cephalosporines (3GC)- and carbapenems-resistant Gram-negative bacteria on organic fresh vegetables” What is the meaning of this sentence?

Expanded espectrum detalactamases (ESRLs) ? Are these ESBLs ?

Low resistance levels were found to ceftazidime and meropenem. How low level resistance is defined ? No MIC is performed in methodology?

No pathogenic bacteria was isolated. How can the authors say that no pathogenic bacteria ? A. baumannii, Pseudomonas aeruginosa and Enterobacter are not pathogenic ?

PCR assays revealed blaVIM to be the most frequently isolated ESBL. blaVIM is more appropriate to classify as Carbapenemases

In abstract the authors have mentioned “A total of 161 strains were isolated”. In methodology “A total of 673 presumptive 3GC- and carbapenems-resistant isolates were obtained (20% 114 from lettuce, 23% from spinach, 40% from cabbage and 36,6% from strawberries), being 74% oxidase-positive. One hundred and sixty-one out of these 637 isolates were randomly 116 chosen for further analysis

Overall, the manuscript have a huge number of such mistakes. No novelty was found in this work. NO MIC was performed and resulted are only based on the disc diffusion method.

Author Response

Thak you

Reviewer 3 Report

The Authors aimed to discuss the Prevalence and characterization of beta-lactam and carbapenems resistant bacteria isolated from organic raw vegetables retailed in Eastern Spain. The manuscript is delivering some interesting pieces of information and is written in a good manner. The manuscript can be accepted after final language and grammar corrections.

Author Response

Thank you

Round 2

Reviewer 1 Report

After reviewing the revised manuscript and the response letter, I have come to the conclusion that the authors made comprehensive modifications following my suggestions and comments , and the original version of the manuscript has been very much improved, and thus I recommend to accept the manuscript for publication

Reviewer 2 Report

The authors have significantly revised the manuscript and therefore can be accepted for publication